# Expanded Graphite/Paraffin/Silicone Rubber as High Temperature Form-stabilized Phase Change Materials for Thermal Energy Storage and Thermal Interface Materials

**DOI:** 10.3390/ma13040894

**Published:** 2020-02-17

**Authors:** Yafang Zhang, Wang Li, Juhua Huang, Ming Cao, Guoping Du

**Affiliations:** 1School of Mechatronics Engineering, Nanchang University, Nanchang 330031, China; zhangyafang@email.ncu.edu.cn (Y.Z.);; 2The Center of Collaboration and Innovation, Jiangxi University of Technology, Nanchang 330098, China; 3School of Materials Science and Engineering, Nanchang University, Nanchang 330031, China

**Keywords:** phase change material, expanded graphite, paraffin, silicon rubber, composite

## Abstract

In this work, expanded graphite/paraffin/silicone rubber composite phase-change materials (PCMs) were prepared by blending the expanded graphite (EG), paraffin wax (PW) and silicone rubber (SR) matrix. It has been shown that PW fully penetrates into the three dimensional (3D) pores of EG to form the EG/PW particles, which are sealed by SR and evenly embedded in the SR matrix. As a result of the excellent thermal stability of SR and the capillary force from the 3D pores of EG, the EG/PW/SR PCMs are found to have good shape stability and high reliability. After being baked in an oven at 150 °C for 24 h, the shape of the EG/PW/SR PCMs is virtually unchanged, and their weight loss and latent heat drop are only 7.91 wt % and 11.3 J/g, respectively. The latent heat of the EG/PW/SR composites can reach up to 43.6 and 41.8 J/g for the melting and crystallizing processes, respectively. The super cooling of PW decreased from 4.2 to 2.4 due to the heterogeneous nucleation on the large surface of EG and the sealing effect of the SR. Meanwhile, the thermal conductivity of the EG/PW/SR PCMs reaches 0.56 W·m^−1^·K^−1^, which is about 2.8 times and 3.73 times of pure PW and pristine SR, respectively. The novel EG/PW/SR PCMs with superior shape and thermal stabilities will have a potential application in heat energy storage and thermal interface materials (TIM) for electronic devices.

## 1. Introduction

The environmental pollution issues and the exhausting resources of petrochemical fuels have accelerated the development of clean and sustainable energy as well as energy storage technologies. One of them is the thermal energy storage technology. Thermal energy that can be directly used and easily assembled has attracted much attention [1,2]. One important material for thermal energy storage is a phase-change material (PCM), which can absorb and release a large amount of thermal energy during the phase change process (solid–liquid, solid–solid and liquid–gas). Due to their high energy storage density and nearly isothermal operating characteristic, PCMs have been widely used in solar energy, industrial waste heat recover, building and battery management [3,4,5]. According to the different working modes and structures, the systems of PCM in solar energy and industrial waste heat recovery can be divided into two types, one is heat pipe heat exchanger and the other is the regenerative phase change energy storage system. During working, PCM absorbs heat and then transmits the heat to the required places through the heat exchange medium, such as power generation, heating water and cooking. Desirable PCMs should have large thermal storage capacity and high thermal conductivity. The latter ensures that heat, to be absorbed or released, can be quickly transferred within the PCMs.

Among many types of PCMs, PW is the most preferred one because of its high energy density, nontoxicity and low vapor pressures. However, the drawback of PW is low thermal conductivity and the possibility of liquid leakage during the phase change process, which restricts its broad applications [6,7]. In order to increase its thermal conductivity, several thermally conductive fillers, such as ceramic fillers Al_2_O_3_ [8,9], AlN [10] and metal nanoparticles Cu [11], Al [12] and Ag [13], were added into PW. For its liquid leakage issue, there are two approaches have been employed. One approach is to use porous materials including diatomite [14], EG [15] and carbon nanotube [16] to absorb PW for preventing its leakage. The other is to microencapsulate PW into core/shell structures. Various types of core and shell materials have been widely explored, including graphene [17,18], polymer shells of PMMA [19], PS [20] and PU [21], and inorganic shells of SiO_2_ [4,22,23,24,25] and CaCO_3_ [26], etc. In general, the liquid leakage issue of PW can be effectively prevented by utilizing the compatibility between a polymer and PW, as the polymeric matrix can fix PW by a strong intermolecular force, therefore suppressing PW leaching [27,28]. Sari et al. [27] blended PW with high density polyethylene (HDPE) to obtain form-stable PCMs. The HDPE and PW have similar chemical and structural characteristics, and this therefore ensures a good compatibility between the two components. Lian et al. [29] prepared a form-stable PCM consisting of PW and epoxy resin, in which the epoxy resin was firstly grafted with adecanethiol (ODT) as ODT and PW have good compatibility. Gao et al. [30] also fabricated a form-stable PCM via blending PW, SiO_2_-EG-PW and HTPB polybutadiene, in which EG was added to improve the compatibility between PW and HTPB. They found that the SiO_2_-EG-PW/PW/HTPB composites can sustain its form stability at 120 °C for 1 h.

Recently, PW-based PCMs have been widely used in battery management and electronic devices to control their temperature for improving their safety performance and service life [31,32,33,34,35]. Wu et al. [32] reported that the battery thermal management with EG/PW PCMs displayed a much better cooling effect than that with natural air cooling. Rajesh et al. [34] filled PW into an aluminum heat sink for electronic equipment cooling, and they found that their operational performance could be significantly improved.

It is worth to note that these polymers currently used in the PCMs usually cannot be used at relatively high temperatures for a long time due to their low thermal stability. This largely limits their applications to a large extent. However, SR has a unique adaptability to extreme temperatures, good elasticity and excellent chemical inert properties. SR has therefore been used as thermal interface materials to reduce the thermal interface resistance for more efficient heat dissipation and increase of span life [36,37,38]. Similarly, improved performance would be expected if SR is used in PCMs [39,40,41]. Fillers with high thermal conductivity, such as graphene, have been utilized in SR based PCMs to enhance their conductivity [40]. However, the published research generally requires the use of microencapsules of PW to avoid the leakage issue [39,40,41]. This considerably complicates the fabrication process of SR based PCMs.

Herein, instead of using the complex microencapsulation procedure and expensive material as graphene, we intend to use more cheaper material EG to capture PW and simultaneously enhance the high thermal conductivity of SR based PCMs. A new PCM composite with large thermal storage capacity and high thermal conductivity was fabricated using PW, EG and SR as the raw materials. The EG/PW particles were firstly prepared, and they were then dispersed into SR by mechanical mixing to form the EG/PW/SR PCMs. The EG not only improves the compatibility between PW and SR, but also enhances the thermal conductivity of the EG/PW/SR PCMs. The SR wraps the EG/PW particles tightly to ensure excellent form-stability of the EG/PW/SR PCMs. The chemical structure, phase change temperature, phase change enthalpy, thermal conductivity and thermal stability of the EG/PW/SR PCMs were systematically studied.

## 2. Experimental

### 2.1. Materials

EG (80 mesh) was supplied by Qingdao Xingtanyuan Company. PW (Tm = 52–54 °C) was purchased from Shanghai Aladdin Company (Shanghai, China). Polymethyl Hydrosiloxane (PMHS, viscosity 20 mm^2^/s, content of hydroxyl group 1.59 wt %), polydimethylsiloxane vinyl terminated (Vi-PDMS, viscosity 500 mm^2^/s; content of vinyl group 0.43 wt %) and platinum catalysis (Pt, 2000 ppm) were purchased from Shenzhen JiPeng Material Company (Shenzhen, China).

### 2.2. Preparation of EG/PW PCM

PW was melted at 80 °C in a glass beaker, and EG (EG:PW = 10:90 in weight) was added into the molten PW solution under continuous stirring. After all of the molten PW was absorbed by the EG, the mixture was cooled to room temperature, and the EG/PW PCM powder was obtained.

### 2.3. Preparation of SR and EG/PW/SR Composite

The neat SR was prepared using the bi-component liquid (Vi-PDMS and PMHS) method with the Vi-PDMS as basic gum, the PMHS as a crosslink agent, and platinum (Pt) as a catalyst. In this work, the molar ratio of the vinyl group and hydroxyl group was controlled at 1:1.2. The detailed preparation process of SR was as follows. Vi-PDMS solution was mixed with PMHS solution at a weight ratio of 100:1.2 under stirring. Platinum catalysts were added to accelerate the reaction. After that, the mixture was placed in a vacuumed glass container for 10 min, and the mixture was poured into a homemade mold for curing in a dry oven at 80 °C for 2 h to obtain the SR. For preparing the EG/PW/SR composite, the EG/PW powder with different contents (10 wt %, 20 wt % and 30 wt %) was added into bi-component liquid. In this paper, The EG/PW/SR composites obtained with different EG/PW contents of 10 wt %, 20 wt % and 30 wt % were denoted as PCM-1, PCM-2 and PCM-3, respectively. The synthesis process of the EG/PW/SR composite is shown in Figure 1.

### 2.4. Characterization

The morphology and microstructure of PW, EG/PW and EG/PW/SR were observed on a field emission scanning electron microscopy (FESEM, JEOL, JSM-6701F, Tokyo, Japan). The chemical composition of PW, EG/PW and EG/PW/SR was tested by Nicolet 5700 Fourier transform infrared (FTIR, Thermo Fisher Scientific, Waltham, MA, USA) spectroscope with a KBr sampling sheet. The wavenumber range is 500–4000 cm^−1^. The crystalline structure of the materials was examined by X-ray diffraction XRD (D8 ADVANCE Bruker, Karlsruhe, Germany). A thermal stability test was performed by a thermal gravimetric analyzer (TGA4000, PE, Waltham, MA, USA) with the temperature range of 30–700 °C at a heating rate of 10 °C/min in the argon atmosphere. Differential scanning calorimetry (DSC) was carried on DSC (DSC8000, PE, Waltham, MA, USA) with materials heated and cooled at a rate of 5 °C/min between 0 and 75 °C in the argon atmosphere. The thermal conductivity coefficient of samples was measured by transient hot wire method at 25 °C (XIATECH, TC3000, Xi’an, China) with a sample dimension of Φ 40 mm × 10 mm. For the form-stability performance, the samples were placed into a drying oven at 150 °C for 24 h to observe the shape stability property and measure the weight loss at a specified time of 0, 1, 2, 12 and 24 h, respectively.

## 3. Results and Discussion

### 3.1. Morphology of the EG/PW/SR Composites

Figure 2 shows morphologies of the EG, EG/PW, SR and EG/PW/SR composite. The EG had a porous wormlike structure (Figure 2a), and it contained many irregular and reticulate pores on the surface of EG, which are consistent with the previous reports of EG [30]. As shown in Figure 2b, a large number of PW filled into the microporous EG, and meanwhile a part of the paraffin was exposed outside of the pores, just coating on the surface of EG. The pure SR had a smooth and flat surface, and no pores and bubbles were observed (Figure 2c). It is proved that the silicone rubber obtained by addition reaction of 1:1.2 molar ratio of vinyl to the hydrogen group is available. (Figure 2d) shows the micrographs of the prepared PCM-1. There were no pores on the fractured surface of PCM-1. It can be observed that EG/PW particles was homogeneously distributed and embedded in the SR matrix. The well dispersed EG should benefit for forming a heat conduction channel and consequently improving the thermal conductivity. The SEM photograph of PCM-2 is shown in Figure 2e. The EG/PW particles were still embedded in the SR matrix. Additionally, it is obviously seen that the body of EG/PW particles was tightly coated by SR, but the sharp edge of EG seemed not to be coated (see the expanded view of Figure 2e). However, a few numbers of EG/PW particles that disembedded into the SR matrix was observed on the fracture surface of PCM-3, as shown in Figure 2f. This phenomenon is probably due to the increased content of EG/PW that gradually deteriorates the mechanical properties of the composite materials. As we know, the PW has poor mechanical properties and it acts as a lubrication effect, which may decrease the binding force between EG particles and the SR matrix [27]. Although the EG/PW particles was disembedded form the matrix, the surface of EG/PW was still wrapped by a thin lay of SR, which lead to sealing the surface of EG/PW. Therefore, the presence microstructure of EG/PW/SR PCM could potentially support PW possessing high temperature form stability properties.

### 3.2. Chemical Structure of EG/PW/SR Composites

The FTIR spectra of the PW, EG/PW and EG/PW/SR composite are displayed in Figure 3. For the spectrum of EG, there was only one broad absorption peak at 3450 cm^−1^, which could be attributed to the stretching vibration of the O–H group due to a little H_2_O absorbed [16]. For the pristine SR, the broad band at 1015–1085 cm^−1^ and 804 cm^−1^ belong to the asymmetric stretching of Si–O–Si and symmetric of Si–O–Si, respectively, which represents the characteristic skeleton of silicone rubber [42]. The Si–O bond is difficult to be broken because it has strong polarity and high bond energy, which can reach as high as 462 KJ/mol [43]. Therefore, the silicone rubber has a good thermal stability that can be a good supporting material for PW. As for pure PW, the absorption peaks can be seen at 2918 cm^−1^ (–CH_2_ symmetric stretching vibration), 2849 cm^−1^ (–CH_3_ symmetric stretching vibration), 1466 cm^−1^ (–CH_2_ bending vibration) and 722 cm^−1^ (–CH_2_ rocking vibration) [4,23]. For the EG/PW/SR composite, all of the corresponding characteristic peaks of PW and SR are presented, without appearing new peaks. These results indicate that the good compatibilities between EG/PW particles and SR matrix.

### 3.3. Crystalline Characterization of EG/PW/SR Composites

To investigate the crystalline property of the EG/PW/SR composite, an XRD was performed and the results are presented in Figure 4. As shown in Figure 4, PW exhibited two sharp diffraction peaks located at 21.3° and 23.1°, corresponding to the (110) and (200) crystal plane [22]. For EG, there is only one diffraction peak at 26.7°, corresponding to (002) crystal plane of graphitic. A broad diffuse peak at around 12.1° is ascribed to the silicone rubber. It demonstrates that the SR possessed the crystalline characteristics of short-range order and long-range disorder. With the increased addition of EG/PW particles, the broad diffuse peak intensity is enhanced, which indicates that the addition of EG/PW promotes SR nucleation and growth. This phenomenon is probably because the EG has a larger specific surface area, which can promote non-uniform nucleation of SR on the surface of EG. These results are consistent with the observation of SEM. It is noted that all of the diffraction peaks exist in the EG/PW/SR composite and no other new peaks are detected. The results of XRD are also consistent with FTIR, which demonstrates that three different component in EG/PW/SR have good compatibility.

### 3.4. Phase Change Properties of EG/PW/SR Composites

The DSC instrument was carried out to study the thermal properties of PW, EG/PW and EG/PW/SR composites. The results are shown in Figure 5 and Table 1. As shown in DSC cures, it can be observed two phase change peaks on the both heating and cooling cures. The minor peak corresponds to the solid–solid phase change of PW and the other peak at a temperature range of 48–55 °C corresponds to the solid–liquid phase change of PW [44]. The phase change parameters of PW and EG/PW/SR composites are presented in Table 1. The T_m_ and T_c_ are peak temperatures during the melting and crystalline process. The super-cooling (ΔT) of PW, EG/PW and EG/PW/SR are listed in Table 1. Super-cooling is the phenomenon that the liquid phase change material can not nucleate until the temperature is below the melting temperature. As shown in Table 1, the super-cooling of PCM-1, PCM-2 and PCM-3 are 1.7, 2.5 and 2.4 °C, respectively, which are lower than that 4.7 °C of PW and 3.4 °C of EG/PW. This decline of super-cooling is ascribed to the heterogeneous nucleation on the large surface of EG and partly on the surface of SR, which also proved that SR was tightly coated on the EG/PW particle surface. Apparently, the decrease of super-cooling could accelerate the phase change cycle and improve the utilization efficiency of the PCM. Besides, the measured latent heat (ΔH) of PCM-1, PCM-2 and PCM-3 were 14.3, 28.5 and 42.7 J/g, respectively. As the content of EG/PW increased, the latent heat of EG/PW/SR increased greatly because the mass fraction of the PW in the composites increased. However, the theoretical value of enthalpy (ΔH^T^) was higher than the measured enthalpy (ΔH). The major reason may be the loss of PW during the curing process (80 °C, 2 h). The other reason may be a little of PW dissolved into the silicon matrix during the addition reaction of the SR synthesis process.

### 3.5. Thermal Stability of EG/PW/SR Composites

The thermal gravimetric analysis (TGA) was used to measure the thermal decomposition temperature of PCM. Figure 6 shows TGA cures for PW, SR, EG/PW and EG/PW/SR composites. It is clearly illustrated that the onset of degradation for pure PW was at about of 145 °C, and the weight loss ended at about 390 °C. For pristine SR, there was a weight loss stage from 370 to 650 °C, due to the degradation of the C–C, C–H, Si–C side chain and Si–O–Si main chain of silicone rubber [43,45]. Compared with PW, SR has higher thermal stability, indicating that SR has the potential to be supporting materials for PW. As for the EG/PW/SR composites, we could observe a two-step decomposition process. The first mass loss step was mainly attributed to the decomposition of PW, while the second decomposition process corresponded to the SR. The onset of degradation of PW in the composites was approximately at 192 °C, 190 °C and 189 °C for PCM-1, PCM-2 and PCM-3, respectively, which indicates that the EG/PW/SR composite PCM can delay the degradation of PW due to the good thermal stability of SR and capillary force provided by the porous structure of EG.

### 3.6. Form stability and Thermal Reliability of EG/PW/SR Composites

Form stability properties of SR, PW, EG/PW and PCM-3 were tested by putting samples into a heated oven at 150 °C for 24 h. In order to prevent the melting PW flowing out, PW was placed into a glass culture dish and the other samples were placed in filter paper. Figure 7 shows the optical photos of PW, SR, EG/PW and PCM-3. The PW melted completely after 1 h, while the shape of PCM-3 had no virtual change after 24 h. The corresponding weight loss chart is shown in (Figure 8a). SR had high thermal stability properties with little weight loss. Compared with other samples, PW had the largest weight loss (57%). Although PW will not leak in the glass dish, the high temperature 150 °C exceeded the decomposition temperature of paraffin, making paraffin oxidized and decomposed in the air. The measured weight loss data of the samples were 36.4% and 7.9% for EG/PW and PCM-3, respectively. The results indicating that the EG/PW/SR composite PCM has a good high temperature form stability. XRD and DSC were performed to test the crystal structure and thermal properties of PCM-3. From (Figure 8b), it can be noted that the XRD patterns of PCM-3 (before baking) and PCM-3 (after baking) were almost identical coupling, but the intensity diffraction peak of PW in PCM-3 after baking slightly decreased compared with that of PCM-3 before baking. No new phase peaks appeared, which indicated that EG/PW and SR have good compatibility. (Figure 8c) shows the DSC curve of PCM-3 before baking and after baking, and the relevant data were presented in Table 2. After baking, the latent heat value of PCM-3 decreased by 11.3 J/g, but the latent heat was still as high as 31.4 J/g. It is interestingly noted that the melting peak temperature T_m_ and crystallizing peak temperature T_c_ were both shifted toward higher temperature. The increasing of T_m_ may be because the melting PW penetrated into the smaller pores of EG during the curing process (150 °C, 24 h), and the pore size also influenced on the melting temperature. As described by the Gibbs–Thomson equation, $∆K\propto \frac{1}{r-t}\;$, where ∆K is the melted temperature change, r and t are the pore radius and thickness of PCM respectively. r is inversely proportional to ∆K, hence, the smaller the pore size will lead to a higher melting temperature [46]. The increased T_c_ may be because the baking caused some PW decomposition and the residual of the PW can be more easily contacted with EG surface. The EG surface can benefit for PW heterogeneous nucleation. Figure 8d shows the DSC curves during 20 thermal cycles of PCM-3 (after baking). It is obviously viewed that the DSC curves of 20 thermal cycles fully coincide, indicating that the sample has excellent circulation stability.

### 3.7. Heat Conduction Test

For phase change materials, heat conduction is an important performance parameter, which reflects the heat storage and heat release efficiency of the material. Figure 9 shows that the heat conductivity of pure paraffin and silicone rubber was only 0.2 W·m^−1^·K^−1^ and 0.15 W·m^−1^·K^−1^, respectively. After adding EG/PW, the thermal conductivity of composite phase change materials increased, and the thermal conductivity of PCM-3 reached a maximum value of 0.56 W·m^−1^·K^−1^, which was 2.8 times that of pure paraffin and 3.73 times that of pure SR. This is because EG has good thermal conductivity and disperses well in the silicon matrix.

## 4. Conclusions

In this study, high temperature form-stable EG/PW/SR composite PCMs were prepared by simply blending EG, PW and SR together. The SEM and XRD analysis proved that the surface of EG/PW was sealed by SR. The DSC test shows the fusion latent heat and the crystallization latent heat of the composite PCM was 43.6 J/g and 41.8 J/g, respectively. The shape stable test indicates that the EG/PW/SR composite PCM could withstand baking at 150 °C for 24 h with no virtual shape change. The XRD crystal structure analysis and DSC thermal cycle test proved the composite PCM had good compatibility and good thermal reliability. The thermal conductivity of the composite PCM could reach a maximum value of 0.56 W·m^−1^·K^−1^, which was 2.8 times that of pure PW and 3.73 times that of pristine SR, respectively. Therefore, the novel EG/PW/SR PCM with superior shape stability and thermal stabilities should have a potential application in heat energy storage, battery management and thermal interface materials for electronic devices. Especially, the new PCM with the facilitate prepared process and low cost can accelerate it popularization and application, while the unique synergism encapsulation mechanism may help for preparing other high temperature form-stabilized PCM.

## Figures and Tables

**Figure 1 materials-13-00894-f001:**
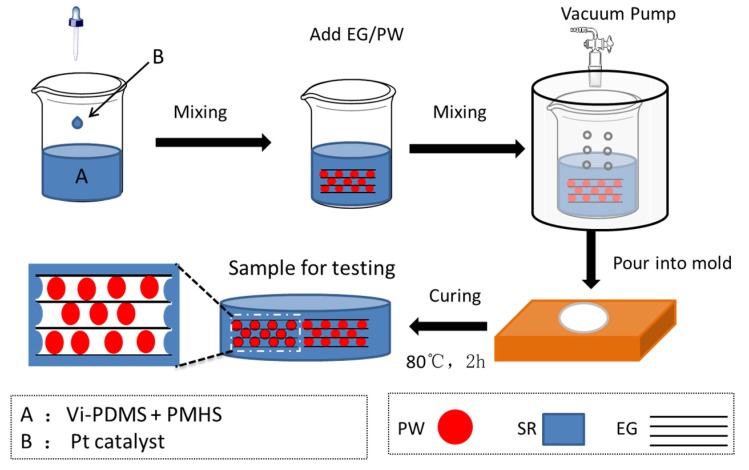
Schematic illustration of the preparation process of expanded graphite (EG)/paraffin wax (PW)/silicone rubber (SR) composites.

**Figure 2 materials-13-00894-f002:**
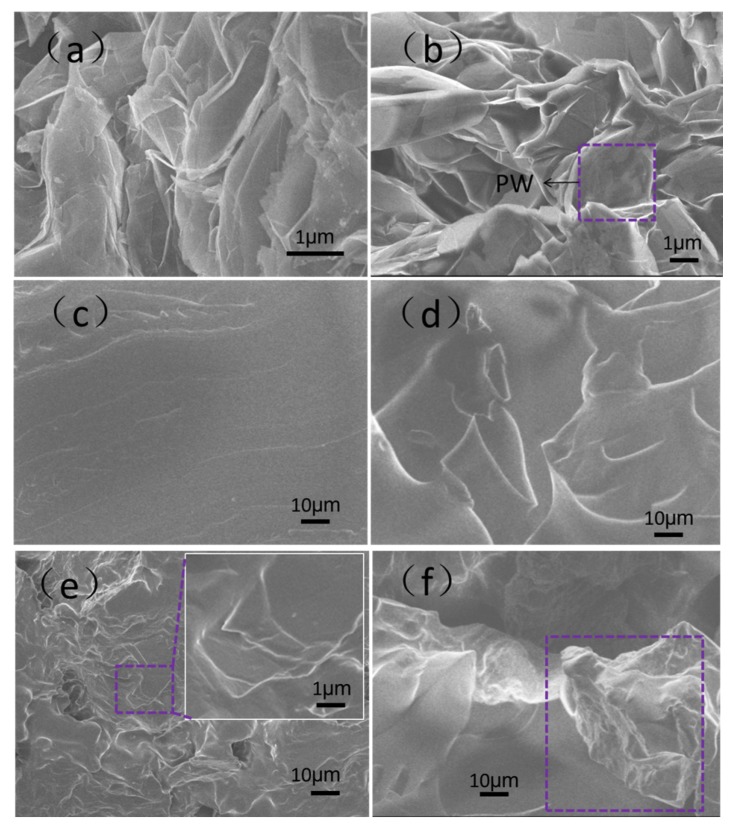
SEM micrographs of (**a**) EG, (**b**) EG/PW, (**c**) SR, (**d**) phase-change material (PCM)-1, (**e**) PCM-2 and (**f**) PCM-3.

**Figure 3 materials-13-00894-f003:**
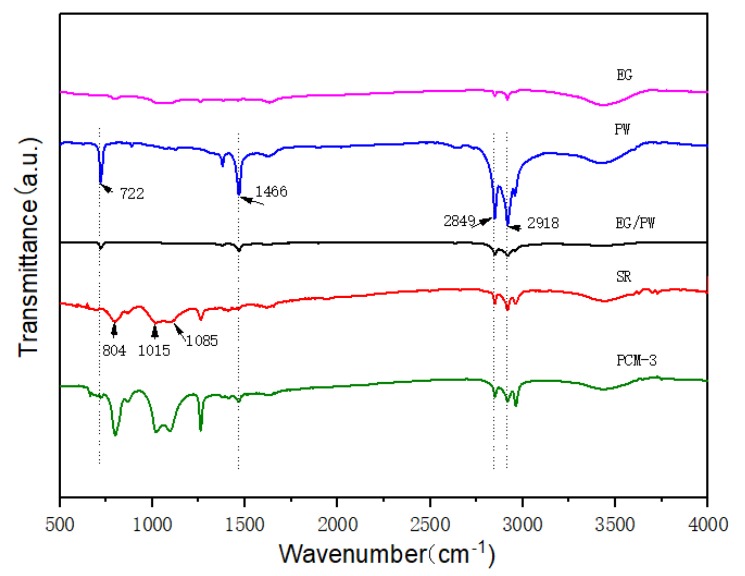
FTIR cures of EG, PW, EG/PW, SR and PCM-3.

**Figure 4 materials-13-00894-f004:**
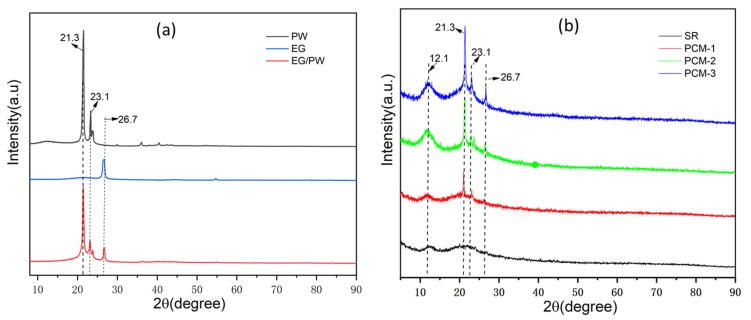
(**a**) XRD diagrams of PW, EG and EG/PW and (**b**) XRD diagrams of SR and as obtained EG/PW/SR composites.

**Figure 5 materials-13-00894-f005:**
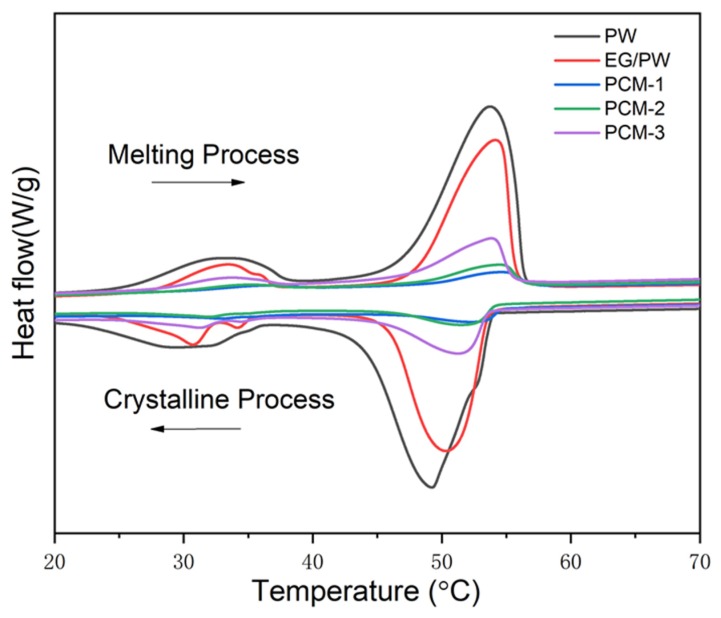
DSC cures of the pure PW, EG/PW and EG/PW/SR composite.

**Figure 6 materials-13-00894-f006:**
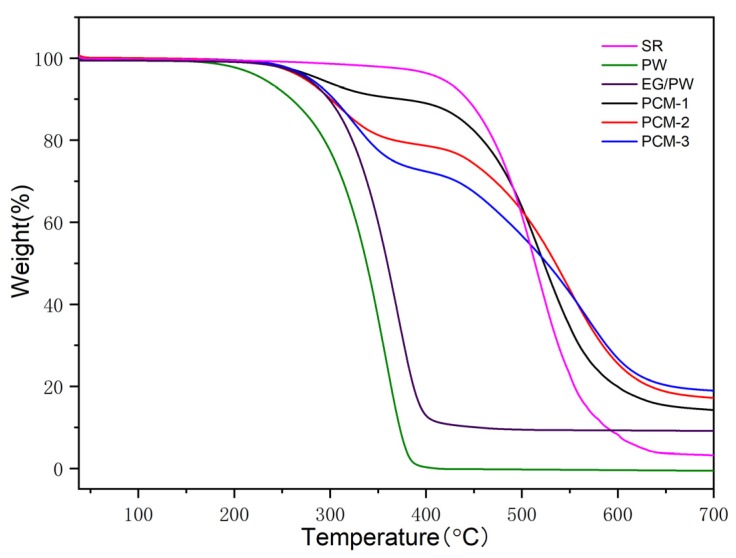
TGA cures of SR, PW, EG/PW and EG/PW/SR composites.

**Figure 7 materials-13-00894-f007:**
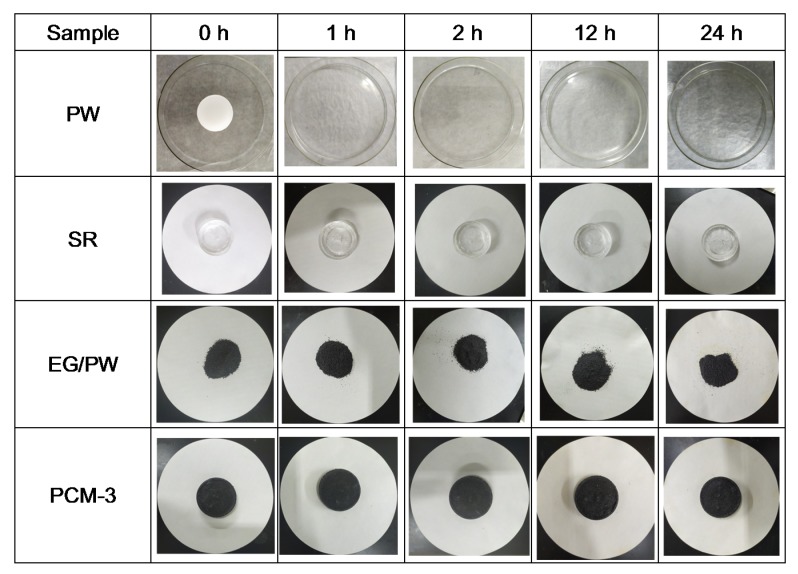
Optical photographs of PW, SR, EG/PW and PCM-3 deposited at 150 °C for 24 h.

**Figure 8 materials-13-00894-f008:**
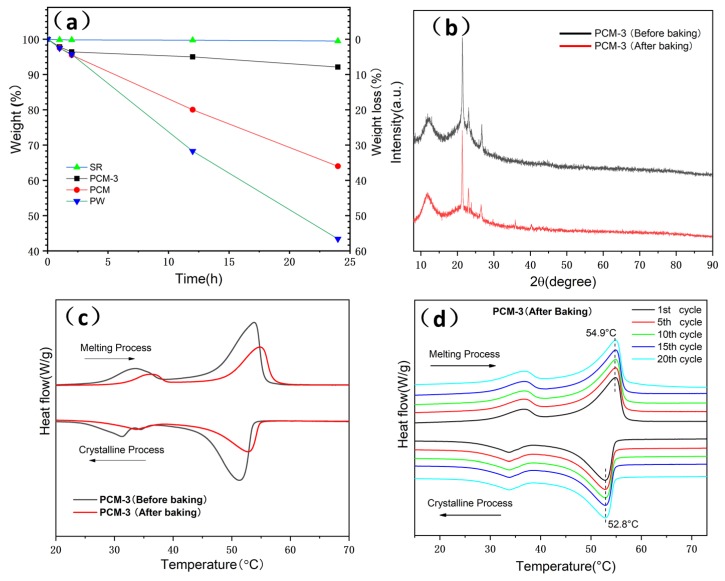
(**a**) Thermal stability analysis of SR, PW, EG/PW and PCM-3, (**b**) XRD spectra and (**c**) DSC curves spectra of PCM-3 before and after (150 °C, 24 h) baking and (**d**) DSC curves of PCM-3 (after baking) with in 20 cycles. Weight loss = $\frac{M_{0}-M_{t}}{M_{t}}\times 100\%$, where M_0_ is the initial mass of samples and M_t_ is the mass of samples weighted at specific time.

**Figure 9 materials-13-00894-f009:**
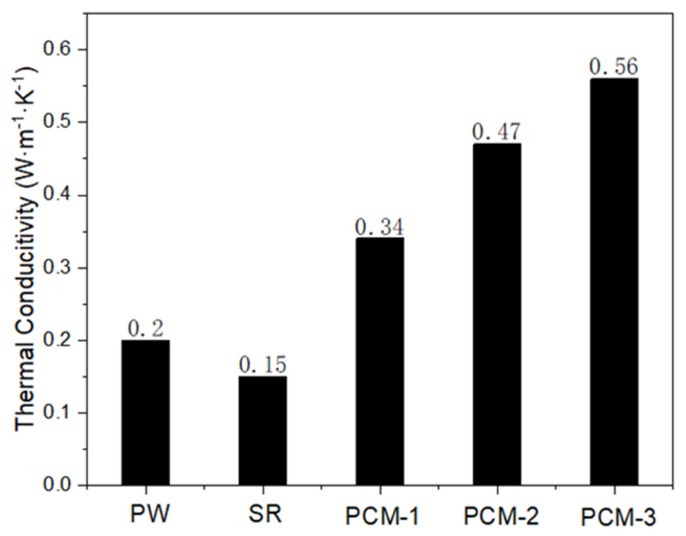
Thermal conductivity coefficients of PW, SR and EG/PW/SR composites.

**Table 1 materials-13-00894-t001:** Thermal characteristics of the PW, EG/PW and EG/PW/SR composite.

Samples	Melting Process		Crystalline Process		-	-	-	-
	T_m_ (°C)	ΔH_m_ (J/g)	T_c_ (°C)	ΔH_c_ (J/g)	ΔT (°C)	ΔH (J/g)	ΔH^T^ (J/g)	ΔH_loss_ (%)
PW	53.8	172.3	49.1	170.1	4.7	171.2	-	-
EG/PW	53.9	149.9	50.5	146.2	3.4	148.1	154.1	3.91
PCM-1	54.4	14.9	52.7	13.7	1.7	14.3	15.4	7.19
PCM-2	54.3	28.7	51.8	28.3	2.5	28.5	30.8	7.51
PCM-3	53.9	43.6	51.5	41.8	2.4	42.7	46.2	7.58

T_m_ is the melting temperature, T_c_ is the crystalline temperature, ΔH_m_ is the melting latent heat, ΔH_c_ is the crystalline latent heat, ΔT = T_m_ − T_c_, ΔH = (ΔH_m_ + ΔH_C_)/2, ΔH^T^ = m_PW_/M_0_ × ΔH_PW_, ΔH_loss_ = (ΔH^T^ − ΔH)/ΔH^T^, where ΔT is supercooling, m_PW_ is the mass of PW, M_0_ is the mass of initial PCM composite, ΔH_PW_ is the ΔH of PW, ΔH^T^ was calculated by ΔH_PW_ multiplying the weight percentage of PW.

**Table 2 materials-13-00894-t002:** Thermal characteristics of PCM-3 before and after baking.

Samples	Melting Process		Crystalline Process		-	-	-
	T_m_ (°C)	ΔH_m_ (J/g)	T_c_ (°C)	ΔH_c_ (J/g)	ΔT (°C)	ΔH (J/g)	ΔH Decrease (J/g)
PCM-3(Before baking)	53.9	43.6	51.5	41.8	2.4	42.7	-
PCM-3(After baking)	54.9	31.8	52.8	30.9	2.1	31.4	11.3

The parameter ΔH decrease = 42.7(J/g) − 31.4(J/g) = 11.3 (J/g).

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
