# Peer review of "Expanded Graphite/Paraffin/Silicone Rubber as High Temperature Form-stabilized Phase Change Materials for Thermal Energy Storage and Thermal Interface Materials"

_materials, 2020, doi:10.3390/ma13040894_

Round 1
Reviewer 1 Report
Summary and overall evaluation
The paper title is “Expanded Graphite/Paraffin/Silicone Rubber as High Temperature Form-stabilized Phase Change Materials for Thermal Energy Storage and Thermal Interface Materials” and generally it is coherent with the described work in the paper.
In the first chapter there are general information: review and introduction. In the second chapter there is materials description and methodology of preparation. The third chapter is very wide and contains different kinds of results. The article has 46 well-chosen references and it is also well written. Reviewer has some remarks which are below.
Specific comments and improvement points
The reviewer doesn’t have remarks to methodology and tests, because in his opinion this part is described enough. However he has one, in his opinion huge major concern. Namely, the authors highlighting in first part of the manuscript that the main problem with PCMs is low conductivity (keeping high latent heat) and they are showing their composite as a respond to that problem. However in reviewer’s opinion the composite has thermal conductivity equals ONLY 0,56 W·m-1·K-1 and 43,6 J/g of latent heat. Of course the authors have right that they increase thermal conductivity 2.8 times of pure PW, but they decrease 4.04 times of latent heat of pure PW! Moreover, comparing the new composite with water in range of 48-55 degrees Celsius (which is the latent temperature of the composite) it is possible to store 43,6 J/g heat in the new composite and 29,3 J/g in water! Taking into account problems with heat transfer in PCMs and difficulty in heat exchanger designs the reviewer thinks that proposing composite is not attractive as a alternative to water. The reviewer as a person who works with 2 big heating systems with PCMs storage tanks thinks that the range of latent temperature is good but he doesn’t see more advantages of the new composite.
Minor concerns:
- l. 85 empty chapter
- in l. 109 is "Scheme 1" but in reviewer's opinion it should be "Fig 1"
- l. 124 empty chapter
Recommendation
Reviewer appreciates the work that has been done by the authors, especially that in the manuscript there has been described a lot of tests. However in his opinion the authors should focus more on final goal, because the final product is not attractive – it is just another PCM, that doesn’t solve problems that trouble that materials.
To sum up, although the article is interesting, it has huge problem, which, according to the reviewer, should be considered and correct advantages should be highlighted, because the authors have chosen incorrect.
Reviewer 2 Report
This work presents several blends of composite materials for thermal energy storage and thermal interface materials. The developed work is explained accurately but further elaboration of the conclusions and significance of this work is needed. Before I can recommend this publication, it is advisable that the authors address the following issues:
-An acronyms table is needed
-The addition of a nomenclature table with the variables employed is encouraged
-Lines 36-37. Elaborate a bit more the uses of PCM in solar energy, waste heat recovery, etc.
-An enhancement of the novelties and significance of this work is needed.
-Line 93: do the authors mean “beaker”?
-Line 106. An additional table with the wt% of PCM-1, PCM-2 and PCM-3 could facilitate the reading
-Line 198: typo: fraction
Line 245: why the smaller pore size leads to higher melting temperatures? Please include this explanation in the text
-Conclusions: The conclusions merely summarize the work done but do not provide information about the significance and impact of the research.
-Line 281: what could be the application of this work to the fields mentioned and why?
Round 2
Reviewer 1 Report
The reviewer agree with the authors about advantages of prepared composite. However in his opinion this composite doesn't have chance to be used in a heating system. So he understand that the manuscript contents information that can be used only in further research as an improvement point.